# Effects of Chitosan Coating with Green Tea Aqueous Extract on Lipid Oxidation and Microbial Growth in Pork Chops during Chilled Storage

**DOI:** 10.3390/foods9060766

**Published:** 2020-06-10

**Authors:** Eduardo Montaño-Sánchez, Brisa del Mar Torres-Martínez, Rey David Vargas-Sánchez, Nelson Huerta-Leidenz, Armida Sánchez-Escalante, María J. Beriain, Gastón R. Torrescano-Urrutia

**Affiliations:** 1Coordinación de Tecnología de Alimentos de Origen Animal (CTAOA), Centro de Investigación en Alimentación y Desarrollo, A.C. (CIAD), Carretera Gustavo Enrique Astiazarán Rosas 46, Hermosillo Sonora 83304, Mexico; eduardo180388@gmail.com (E.M.-S.); brisa.torres@estudiantes.ciad.mx (B.d.M.T.-M.); rey.vargas@ciad.mx (R.D.V.-S.); armida-sanchez@ciad.mx (A.S.-E.); 2Department of Animal and Food Sciences, Texas Tech University, Box 42141, Lubbock, TX 79409-2141, USA; nelson.huerta@ttu.edu; 3ISFOOD, Universidad Pública de Navarra, Campus Arrosadia, 31006 Pamplona, Spain

**Keywords:** chitosan coating, green tea, antioxidant, antimicrobial, meat quality

## Abstract

Lipid oxidation and microbial growth are the major causes of meat quality deterioration. Natural ingredients in meat products have been proposed as a strategy to prevent quality deterioration during cold storage. This study aimed to assess the effects of added chitosan coating, alone and in combination with green tea water extract (GTWE), on the quality of pork chops during prolonged cold storage. For evaluating oxidative and antimicrobial stabilities, 72 fresh pork samples were subjected to four treatments (*n* = 18 per treatment): T0 (non-coated chops without GTWE); T1 (chitosan-coated chops without GTWE); T2 (chitosan-coated chops plus 0.1% of GTWE); and T3 (chitosan-coated chops plus 0.5% of GTWE). Pork samples were stored at 0 °C and subjected to physicochemical evaluation (pH, colour, and lipid oxidation) and microbiological analyses (mesophilic and pyschrotrophic counts) at 0, 5, 10, 15, 20 and 25 days of storage. GTWE presented high total phenolic content (> 500 mg gallic acid equivalents/g); the incorporation of chitosan coatings increased (*p* < 0.05) free radical scavenging activity (FRSA, >90% of inhibition) and microbial growth inhibition (>50% for all tested pathogens), depending on the concentration. Further, GTWE inclusion in pork samples (T2 and T3) reduced (*p* < 0.05) pH, lipid oxidation and microbial counts, as well as colour loss in meat and bone throughout storage. Chitosan coating with GTWE could be used as an additive for the preservation of pork meat products.

## 1. Introduction

Mexico is one of the main pork-producing countries in Latin America. Pig production in Mexico is around 1.4 million metric tonnes, and pork is the second most popular meat item in the country, with a domestic consumption of 2.4 million metric tonnes in 2019. Further, Mexico exports ca. 0.2 million metric tonnes to Asiatic markets like China and Japan [1]. The Mexican pork industry carries out trade with Asian regions via maritime transportation under refrigerated conditions, with delivery times exceeding 20 days [2]. During this relatively long storage period, lipid oxidation (LOX) and microbial growth may occur on board the ship [3].

Water, vitamins and minerals, proteins and fats (particularly saturated fatty acid—SFA, and polyunsaturated fatty acid—PUFA) are the main chemical components of pork [4]. PUFA are involved in the technological quality attributes of pork, such as firmness and colour, as well as in the flavour development during cooking; however, intrinsic and extrinsic factors may adversely affect meat quality in terms of lipid oxidation (LOX) [5,6]. Furthermore, growth of spoilage bacteria reduces shelf-life and jeopardises the palatability of meat and meat products [6]; thus, the growing demand for pork needs to be attended with an adequate control of LOX and microbial growth [5,6].

The most effective way to control LOX is to avoid free radicals’ formation. This can be accomplished by adding synthetic antioxidants (e.g., butylated hydroxytoluene—BHT; and butylated hydroxyanisole—BHA) to the meat product formulation [7]. However, its uncontrolled use in foods may promote adverse effects on human health [8]. Natural antioxidants of plant origin, such as dog roses, ginger, grape, lotus, lemon, oregano, sage, rosemary and green tea, have been proposed to reduce LOX [9]. In addition, shelf-life is increased by reducing the microbial growth using modified atmosphere packaging [10]. There is an increasing interest in the application of edible coatings, like chitosan, in food matrices, motivated not only by the increasing consumer demand for safe and stable food products, but also by the rejection of non-biodegradable packaging [11]. Chitosan has also been proposed to increase the shelf-life of meat products due to its multiple functional properties, including antioxidant and antimicrobial properties [12].

Therefore, the aim of this study was to explore the effects of added chitosan coating, alone and in combination with green tea water extract, on pork quality during prolonged cold storage.

## 2. Materials and Methods

### 2.1. Chemicals and Reagents 

All reagents used were of analytical grade. Chitosan (degree of deacetylation between 75% and 85%, viscosity: 20–800 cPs), Folin–Ciocalteu reagent, sodium carbonate (Na_2_CO_3_), 1,1-diphenyl-2-picrylhydrazyl (DPPH^•^), ethanol, gallic acid, 1,1,3,3-tetramethoxypropane (malonaldehyde), butylated hydroxytoluene (BHT) and ascorbic acid (Asc ac) were purchased from Sigma Chemicals (St. Louis, MO, USA). Trichloroacetic (TCA) and 2-thiobarbituric (TBA) and acetic acids were supplied by J.T. Baker (Baker ^®^, Phillipsburg, NJ, USA), while Brain Heart Infusion broth (BHI) and plate count agar (PCA) were obtained from DifcoTM Laboratories (Detroit, MI, USA). Green tea leaf powder was purchased from REDSA, S.A. de C.V. (CDMX, México).

### 2.2. Extract Preparation

The extraction of the bioactive compounds was carried out via the maceration method [13]. Polyphenols from tea leaf powder were extracted with water at 80 °C as solvent (1:40, *w/v*) over 20 min. Thereafter, the sample was centrifuged (2500× *g*/5 °C/10 min) to collect the supernatant (Beckman, J2-21, Beckman Instruments Inc., USA). The solution was filtered (Whatman 1 filter paper), concentrated under reduced pressure (rotary evaporator BÜCHI R-200, Flawil, Switzerland) and lyophilised with a freeze-dryer (SEDSF12, Samwon Co. Ltd., Busan, Korea). Once the green tea water extract (GTWE) was obtained, it was stored under dark conditions at −20 °C, until further analysis.

### 2.3. Coating Preparation

The chitosan coating (ChC) solutions were prepared as previously described [14]. A chitosan 1% solution (1%, *w*/*v*) was prepared by dissolving 5 g of chitosan in 500 mL of distilled water with glacial acetic acid (1%, *v*/*v*). Afterwards, GTWE was incorporated into the ChC solution and homogenised under magnetic stirring (40 °C for 12 h), until reaching a concentration of 0.1% and 0.5%. The obtained coating-forming dispersions were stored at 5 °C before proceeding to the antioxidant evaluation and pork meat application.

### 2.4. Polyphenols Content and Biological Activity 

#### 2.4.1. Total Phenolic Content

The total phenolic content (TPHC) was determined by the Folin–Ciocalteu reagent assay [15]. An aliquot of GTWE (50 µL, 5 mg/mL), as well as ChC solutions in acetic acid 1% (1:100, *w/v*) incorporated with 0%, 0.1% and 0.5% of GTWE (ChC—0%, ChC—0.1% and ChC—0.5%, respectively), were homogenised with 200 µL of Folin–Ciocalteu reagent (2 M) and 300 µL of Na_2_CO_3_ (7%, *w/v*). The reaction mixture was mixed with 800 µL of distilled water and incubated for 1 h at room temperature (25 °C), under dark conditions. The absorbance was measured at 750 nm in a spectrophotometer (UV-Visible Cary 50 Bio^®^, Varian, CA, USA). The results were expressed as mg of gallic acid equivalents/g of sample (mg GAE/g).

#### 2.4.2. Antiradical Assay 

The free radical scavenging activity (FRSA) was determined by the DPPH method [16]. An aliquot of GTWE, ChC—0.1% and ChC—0.5% (500 µL, 100 µg/mL) was mixed with 500 µL of DPPH^•^ ethanol solution (300 µM) and incubated at 50 °C for 20 min, under dark conditions. Afterwards, the absorbance was measured at 517 nm. Ascorbic acid (Asc ac, 70 µM) was used as antioxidant standard. FRSA was calculated as (Abs A–Abs B/Abs A) × 100, where Abs A is the absorbance of the control (*t* = 0 min), and Abs B is the absorbance of the sample (*t* = 30 min).

#### 2.4.3. Antimicrobial Assay 

The antimicrobial analyses were performed via the liquid nutrient microdilution method against food-borne pathogens [17]. Gram-positive (*Staphylococcus aureus* ATCC 29213B and *Listeria innocua*) and Gram-negative (*Escherichia coli* ATCC 25922 and *Salmonella typhimurium* ATCC 14028) bacteria strains were initially reactivated in liquid nutrient (BHI) broth, and incubated at 37 °C for 24 h. Before testing, all bacteria strains were maintained in glycerol (10%, *v*/*v*) at −20 °C, until used. Afterwards, an aliquot of GTWE, ChC—0.1%, and ChC—0.5% (50 μL, 100 µg/mL) was mixed with 50 μL of the cellular suspension (1.5 × 10^8^ CFU/mL, 0.5 McFarland standard) and incubated at 37 °C for 24 h. Later on, the optical density (OD) was read at 630 nm in a spectrophotometer. Gentamicin (25 µg/mL) was used as the standard, and BHI broth solution as the blank. The inhibition percentage was calculated as (OD630 nm A–OD630 nm B/OD630 nm A) × 100, where OD630 nm A is the absorbance value at 630 nm of the untreated bacteria (*t* = 24 h), and OD630 nm B is the absorbance of the bacteria treated at the same wavelength (*t* = 24 h).

### 2.5. Pork Meat Preparation and Storage

Bone-in pork loins (*Longissimus thoracis* m., fabricated at 24 h *postmortem*, with pH 5.6 and <2 log_10_ CFU/g) were procured from a federally inspected local processing plant. A total of 72 pork chops of 1.5 cm thickness were excised (250 g approximately) to be subjected to the four treatments (3 chops per treatment per day), in two independent experimental trials as follows: T0 (non-coated chops and without GTWE); T1 (chitosan-coated chops without GTWE); T2 (chitosan-coated chops plus 0.1% of GTWE); and T3 (chitosan-coated chops plus 0.5% of GTWE). The samples were kept at 0 °C, chops were submerged in the solution of each treatment for 5 min; and the excess dispersion was drained off. All samples were vacuum-packaged and stored at 0 °C. Three vacuum-packaged samples per treatment were opened for due analysis (with three replications) on each sampling day (0, 5, 10, 15, 20 and 25 days).

### 2.6. Pork Quality Measurements

#### 2.6.1. pH measurement

The samples were homogenised with distilled water (1:10, *w/v*) at 9500 rpm in ice bath at 5 °C for 1 min in a homogeniser (Ultraturrax T25, IKA, Germany). The pH was measured with a potentiometer according to method 981.12 of AOAC (2005) (Model pH 211, Hanna Instruments Inc., Woonsocket, RI, USA) [18].

#### 2.6.2. Colour Measurement

Prior to colour evaluation, the packaging material was removed, and the samples were exposed to atmospheric O_2_ at 0 °C for 30 min under dark conditions, in order to stabilise the colour. Thereafter, 15 measurements on meat and bone surfaces were performed using a spectrophotometer (model CM 508d, Konica Minolta Inc., Tokyo, Japan) with a F2 illuminant and 10° observer [19]. Recorded parameters consisted of lightness (L∗), redness (a∗), yellowness (b∗), Chroma (C∗) and hue angle (h∗).

#### 2.6.3. Lipid Oxidation

Lipid oxidation was determined by the thiobarbituric acid reactive substances (TBARS) formation [20]. Meat samples were homogenised with trichloroacetic acid (10%, *w/v*) in a basic homogeniser (4500 rpm/5 °C/1 min), centrifuged (2500× *g*/5 °C/20 min), and the supernatant was filtered (Whatman 1 filter paper). When meat samples were homogenised, the test tubes were kept in an ice bath at 5 °C to reduce the development of oxidative reactions during extraction of TBARS. Afterwards, 2 mL of 2-TBA solution (20 mm) was mixed with 2 mL of the filtered supernatant, and boiled in a water bath for 20 min. After cooling at 10 °C, the absorbance was measured at 531 nm in a spectrophotometer (Spectrophotometer 336001, Spectronic Genesys 5, Thermo Electron Corp., NY, USA). TBARS values were calculated from a 1, 1, 3, 3–tetramethoxypropane (TMP) standard curve and expressed as mg of malondialdehyde/kg of meat (mg MDA/kg). Three measurement were taken for sample at each sampling day.

#### 2.6.4. Microbial Growth

Mesophilic and psychrotrophic bacterial growth were measured by the pour plate method [21]. Meat samples were homogenised aseptically with peptone water (0.1%, *w/v*) for 1 min using a stomacher (Seward Stomacher^®^ 400, FL, USA); then 1 mL quantities of the appropriate dilutions were pour-plated using PCA as the standard. The inoculated plates were incubated at 37 °C for two days to assess total viable counts of mesophilic bacteria, and at 5 °C for 10 days for psychrotrophic bacteria. All bacterial counts were expressed as the log_10_ of colony forming units/g of sample (log_10_ CFU/g). Three measurement were taken from each sample at each storage time.

### 2.7. Statistical Analysis

Phytochemical antioxidant and antimicrobial data obtained in vitro from the three independent experimental trials (with three replications, were subjected to one-way analysis of variance (ANOVA) with the fixed effect of treatment (GTWE, ChC—0%, ChC—0.1% and ChC—0.5%). Meat quality measurements data obtained from the two independent experimental trials (with three replications) were subjected to a two-way factorial ANOVA, with the antioxidant treatment at four levels (T0, T1, T2 and T3) and storage time at six levels (0, 5, 10, 15, 20 and 25 days) as the main effects, and the two-way interaction. When ANOVA detected a significant effect for any variable response, Tukey HSD tests were carried out for a multiple comparison of means at *p* < 0.05. Mean ± standard deviation values were used as descriptive statistics. In addition, Pearson’s correlation coefficients between the response variables were calculated. All data were analysed using the National Center for Social Statistics statistical software (NCSS, 2007, LLC, Kaysville, UT, USA).

## 3. Results

### 3.1. In vitro Polyphenol Content and Biological Properties

Table 1 reports the in vitro values for the TPHC and biological activity (FRSA and bacterial inhibition) of the GTWE, and chitosan treatments incorporated with GTWE, from 0% to 0.5%. GTWE resulted in the highest TPHC value (>500 mg GAE/g), followed by ChC—0.5% > ChC—0.1% > ChC—0% (*p* < 0.05). In addition, treatments with GTWE and ChC—0.5% displayed the highest (*p* < 0.05) FRSA (> 90% of inhibition), followed by ChC—0.1% (71% of inhibition) > ChC—0% (< 1% of inhibition). High FRSA values were obtained for the standard (Asc ac, >90% of inhibition). A remarkably high (*r*^2^ = 0.990) correlation was detected between TPHC and FRSA among treatments incorporated with GTWE, from 0% to 0.5%.

Moreover, GTWE, ChC—0.1% and ChC—0.5% treatments exhibited the highest inhibitory activity (>90% of inhibition) against *L*. *innocua*, *E*. *coli* and *S*. *typhimurium* (*p* < 0.05), whereas ChC—0.5% (chitosan with 0.5% of GTWE) exhibited the highest (*p* < 0.05) inhibitory activity against *S*. *aureus* (>90%). The results also indicate that the inhibitory effect of chitosan coatings was dependent upon the GTWE concentration. Furthermore, a significantly high (*p* < 0.05) correlation coefficient (*r*^2^ = 0.866) between TPHC and antimicrobial inhibitory activity was detected.

### 3.2. Physicochemical Changes in Pork Samples during Storage Times

As shown in Figure 1, at 0 days of storage, a reduction of pH values (*p* < 0.05) was found in chitosan-treated samples, in comparison to the T0. Although pH values for the control samples increased at subsequent times during storage (*p* < 0.05), no significant differences were found in the pH values (*p* > 0.05) of samples treated with T1, T2 and T3 throughout the whole storage time. At day 25 of storage, the highest (*p* < 0.05) pH values were shown in T0.

The effects of treatment and storage time on the L* (lightness), a* (redness) and b* (yellowness) of the muscle and bone surfaces of the pork chops were significant (*p* < 0.001). At day 0 of storage, L* and b* values were not affected by added coatings (*p* > 0.05), whereas a 30% reduction in the a* values was observed as a result of all coating treatments, when compared to T0 samples (*p* < 0.05). After 25 days of storage, L* and b* values respectively increased 4.0% and 14.8%, in T0 and T1 samples, when compared to the T2 and T3 treatments (*p* < 0.05). Samples treated with T2 and T3 exhibited the highest a* values (27.2%), compared to T0 and T1 (*p* < 0.05).

As shown in Table 2, the bone colour surface analysis of pork chops showed that the initial L* and a* values increased (*p* < 0.05) with chitosan coatings (13.3% and 33.1%, respectively), in comparison with T0. While the b* values were not affected (*p* > 0.05) by chitosan coatings (average value of 16.0) after 25 days of storage, samples treated with T2 and T3 exhibited the lowest L* and the highest a* values, in comparison with T0 and T1 (*p* < 0.05); however, a non-significant effect (*p* > 0.05) was observed in the b* values. 

### 3.3. Lipid Oxidation

The effects of treatment and storage time on the TBARS values were significant (*p* < 0.001). As shown in Figure 2, at day 0 of storage, the TBARS values remained below 0.1 mg MDA/kg for all treatments (*p* > 0.05). Although these values increased during the storage period for T0 and T1 treatments, at the end of the longest storage period, lower TBARS values (*p* < 0.05) were found in the samples from the T2 and T3 treatments (59.1% of TBARS inhibition).

### 3.4. Total Aerobic Bacterial Counts

The effects of treatment and storage time on the mesophilic and psychrotrophic values were significant (*p* < 0.001). As shown in Figure 3, at day 0 of storage, all treatments remained below 2 log_10_ CFU/g (*p* > 0.05), but thereafter these values increased throughout the storage period. However, at the end of the storage period, all coating treatments reduced (*p* < 0.05) the mesophilic and psychrotrophic growth by 49.4% and 41.4% (i.e., 3.2 and 2.5 log_10_ CFU/g, respectively), as compared to T0.

## 4. Discussion

Green tea leaves are known as an important source of metabolites, such as alkaloids, amino acids, glucides, volatile compounds, minerals, and trace elements, as well as polyphenols. Phenolic constituents are the most biologically active group of tea components and are characterised by their containing at least one acidic hydroxyl group bound to an aromatic phenyl ring [22]. Our results confirmed that GTWE is an important source of phenolic constituents with high antiradical activity. These findings agree with previous results regarding the TPHC (200–1900 mg GAE/g) of some commercial GTWE [23]. It has also been reported that green tea extract possesses higher FRSA (>90% of inhibition) in comparison with other herbs extracts, including honeysuckle, jasmine, juhua, lavender, osmanthus, rose, duzhong, lemongrass, mate and rosemary [24]. Our results also indicate that chitosan coatings with 0.1% added GTWE showed the lowest TPHC and FRSA values, which could be influenced by the interaction between the matrix and the antioxidant additives [25].

It is noteworthy that the antibacterial activity of GTWE and coatings was assessed by the broth microdilution method, and tested against *S*. *aureus* and *L*. *innocua* (Gram-positive), as well as *E*. *coli* and *S*. *typhimurium* (Gram-negative), a group of pathogens that are most likely to be found in meat and meat products [26]. Phenolic compounds found in green tea extracts are recognised by their antimicrobial potential against food-borne pathogens, such as *S*. *aureus* and *B*. *cereus* [23]. In agreement with our study, a reduction in *S*. *aureus* and *L*. *monocytogenes* growth (>50%) was reported with GTWE treatment; however, a lower (20%) inhibition of *S*. *enteritidis* and *E*. *coli* was noted [27].

The antimicrobial effectiveness of 1% chitosan coatings [against Gram-positive (*S*. *aureus* and *S*. *epidermis*) and Gram-negative (*K*. *pneumonia* and *P*. *aeruginosa*) bacteria], or that of chitosan coatings mixed with plant extracts (rosemary 1% > olive 1% = capsicum 1%) against *L*. *monocytogenes* growth, has been reported [28,29].

The antimicrobial activity of chitosan against *S*. *aureus* can be associated with the inhibition of nutrient adsorption [30]. While the inhibitory effect against *E. coli* is associated with chitosan’s molecular weight (which, in turn, depends on the concentration of the -NH_2_ groups of the polymer), the deacetylation degree, its concentration in the solution (depending upon -NH^3+^ groups) and the pH of the medium are important modulators [29,30]. In this context, two structural, antibacterial mechanisms have been postulated: (1) chitosan may have bound to the negatively-charged bacterial surface, to disturb the cell membrane and cause cell death; and (2) permeated chitosan oligomers may have blocked the transcription from DNA, and interfered with the RNA and protein [29]. A highly significant correlation exists between TPHC and antioxidant activity in leaf extracts, which indicates that these parameters could be used as quality control criterion for natural extracts [31]. In this regard, the findings obtained from the present work highlight that chitosan coatings incorporated with GTWE are a promising ingredient, and could be applied as a natural additive for extending the shelf life of fresh meats.

The results of chitosan coatings incorporated with 0.1% and 0.5% of GTWE, with regards to meat quality parameters, indicate that the initial (24 h postmortem) pH values of the pork samples were 5.6, which is within the desirable value range [32,33]. A significant reduction in the pH values of meat samples treated with 1% chitosan coatings has been associated with the acid solution employed to dissolve chitosan [34], which agrees with our observations. Regrettably, an increase in pH values has been associated with the utilisation of amino acids by bacteria, and the undesirable production of ammonia and amines [33].

Colour plays an important role in consumer evaluation of meat quality, and is an indicator of freshness [35,36]. Chitosan coatings improved the red colour (11.6%) in pastrima stored at 4 °C for 4 weeks, whereas an insignificant effect was observed in L* and b* values following chitosan treatment, in comparison with the control samples [37]. The meat discolouration process, from a freshly cut pork chop up to the end of its shelf-life, had a relationship with the oxidation process, explaining why a* and b* decreased during oxidation, but changes were more pronounced in the a* values [38]. It has also been suggested that bone surface discolouration is associated with the oxidation process [35,39], and natural antioxidants, such as ascorbic acid, citric acid and other herbs, can reduce colour changes in pork bone during refrigeration storage [39,40].

Our results show that 1% chitosan coatings incorporated with 0.1% and 0.5% of GTWE reduce the lipid oxidation of pork samples. In agreement with this, other researchers [41] found a reduction in MDA formation (33% and >50%, respectively), during storage at 4 °C for 12 days, in pork samples treated with chitosan films and chitosan films incorporated with tea polyphenols. It has also been demonstrated that chitosan coatings (1%), alone or combined with grape seed extracts (0.1%), reduced MDA formation in chicken breast (23% and 62.3%, respectively) during storage at 4 °C for 9 days [42]. Likewise, a reduction in MDA formation (values below 1 mg MDA/kg) has been reported in cooked products (chicken balls, chicken kabab and mutton kabab) incorporated with chitosan coating (1%), stored at 0–3 °C for 14 days [43]. Furthermore, a reduction of MDA formation (48.8%) has been reported in pastirma (a dry-cured meat product) treated with chitosan coating (1.5%) during storage at 5 °C for 4 weeks [37]. Chitosan and GTWE may reduce the initiation of lipid oxidation, and consequently MDA formation, in meat products, by acting as a proton donator or chelator in transition metal ions [23,44].

Our results also show that chitosan coatings incorporated with 0.1% and 0.5% of GTWE reduce microbial growth in pork samples. In agreement with this, a reduction of approximately 2.0 log_10_ CFU/g was observed in mesophilic growth for pork samples treated with chitosan incorporated with tea polyphenols, during storage at 4 °C for 12 days [41]. A reduction of pyschrotrophic growth (1 log_10_ CFU/g) was observed in pork sausages treated with chitosan coating (2%) with the incorporation of an extract containing 0.1% mint (*Mentha spicata*) [43]. Another report [42] found that the mesophilic count in chicken breast meat decreased (1.8 and 2.0 log_10_ CFU/g, respectively) when treated with chitosan coating, alone (1%) or combined with 0.1% grape seed extracts, during storage at 4 °C for 9 days. In addition, a reduction in both mesophilic and pyschrotrophic count (2 log_10_ CFU/g by both) has been reported for chitosan-coated pastirma (1.5%) during storage [37]. Interestingly, a stronger antimicrobial activity has been detected in chitosan when it is dissolved in acid solutions, because under this condition, chitosan acquires a positive charge that interacts with the negatively-charged groups of the bacterial cell wall, such as phosphate, carboxyl, *N*-acetylglucosamine and *N*-acetylmuramic [34,45]. Therefore, in this work a reduction of bacterial growth could also be associated with a decrease in pH values.

## 5. Conclusions

Chitosan coating in combination with GTWE improves the physicochemical (pH, colour and lipid oxidation) and microbiological qualities of bone-in pork chop samples, during the storage times used herein. The inclusion of 0.1% and 0.5% of GTWE in 1% chitosan coatings is effective in delaying MDA formation and microbial growth, and at the same time, it has beneficial effects on the pH and colour of pork meat and bone (a*). These results could be associated with the polyphenolic constituents of the GTWE, as well as with the joint antioxidant and antimicrobial effects of the GTWE and chitosan coating. Our results indicate that a chitosan coating incorporated with GTWE could be used as an additive for the preservation of pork meat products. It would be desirable to carry out consumer studies, to test the results found in the present work.

## Figures and Tables

**Figure 1 foods-09-00766-f001:**
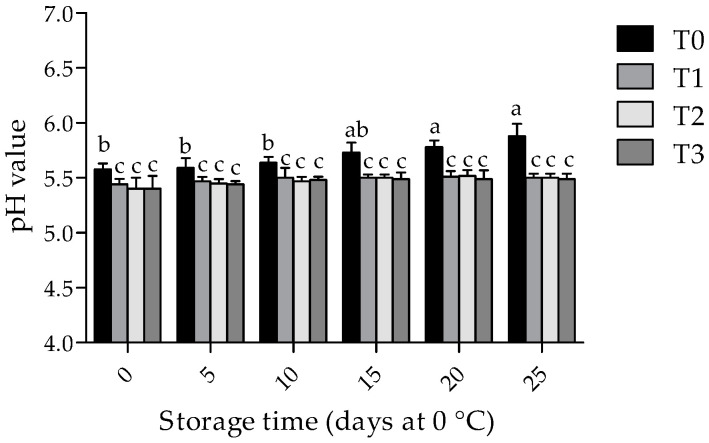
Effect of chitosan with GTWE and storage time on pH values of meat samples. Values expressed as mean ± standard deviation. T0 (non-coated chops and without GTWE); T1 (chitosan-coated chops without GTWE); T2 (chitosan-coated chops plus 0.1% of GTWE); and T3 (chitosan-coated chops plus 0.5% of GTWE). Different superscripts (a–c) indicate significant differences among treatment × storage time values (*p* < 0.05).

**Figure 2 foods-09-00766-f002:**
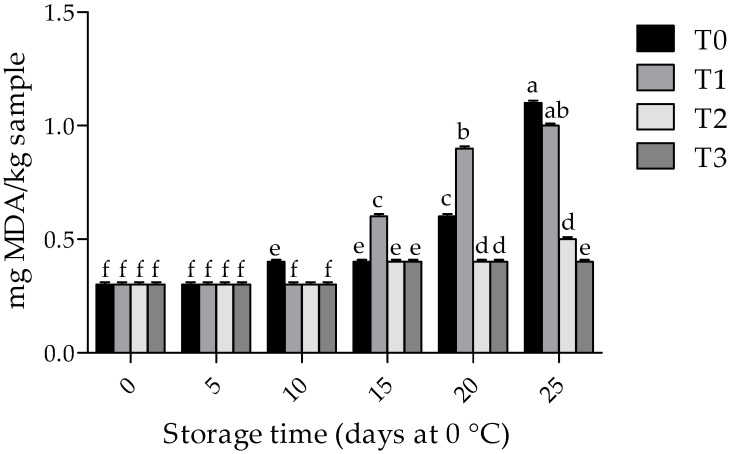
Effect of chitosan with GTWE and storage time on lipid oxidation of meat samples. Values expressed as mean ± standard deviation. T0 (non-coated chops and without antioxidant); T1 (chitosan-coated chops without antioxidant); T2 (chitosan-coated chops with 0.1% of GTWE); and T3 (chitosan-coated chops with 0.5% of GTWE). Different superscripts (a–f) indicate significant differences among treatment x storage time effect (*p* < 0.05).

**Figure 3 foods-09-00766-f003:**
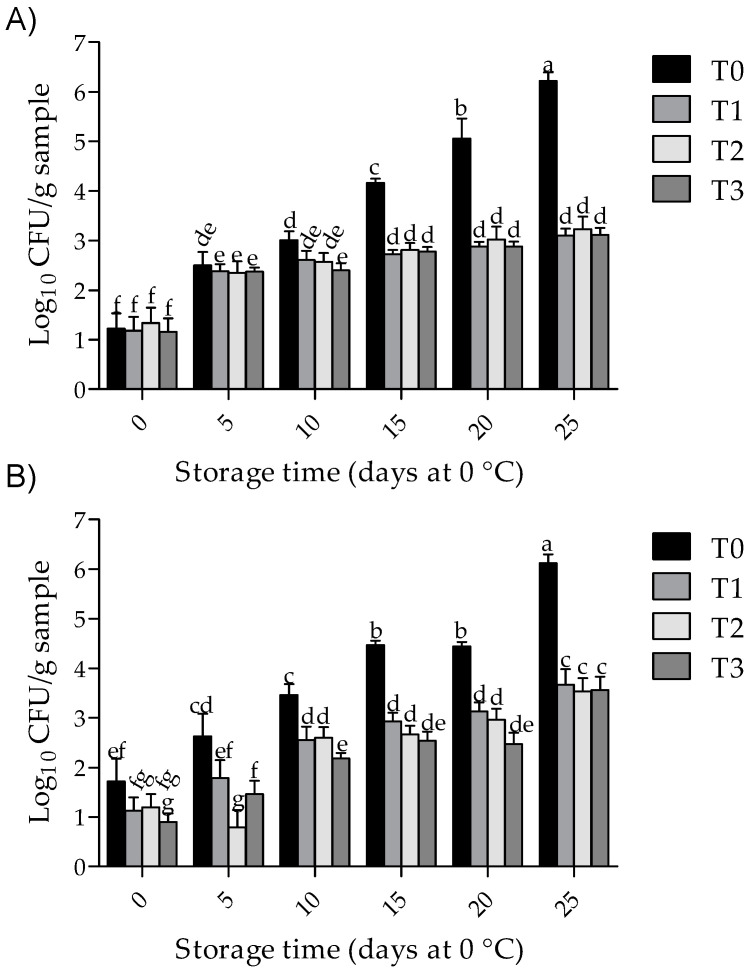
Effect of chitosan with GTWE and storage time on microbial growth of pork chop samples. Mesophilic (**A**) and psychrotrophic counts (**B**); values are expressed as mean ± standard deviation. T0 (non-coated chops and without antioxidant); T1 (chitosan-coated chops without antioxidant); T2 (chitosan-coated chops plus 0.1% of GTWE); and T3 (chitosan-coated chops plus 0.5% of GTWE). Different superscripts (a–g) indicate significant differences among treatment × storage time effects (*p* < 0.05).

**Table 1 foods-09-00766-t001:** In vitro total phenolic content, antiradical and bacterial inhibition of GTWE and chitosan coatings.

Treatment ^1^	TPHC (mg GAE/g)	FRSA (%)	Antimicrobial Assay (%)			
			*S. aureus*	*L. innocua*	*E. coli*	*S. typhimurium*
GTWE	582.7 ± 24.0 ^a^	90.7 ± 1.0 ^a^	86.8 ± 0.2 ^c^	95.1 ± 0.22 ^c^	94.6 ± 0.10 ^d^	91.7 ± 0.17 ^c^
ChC—0%	0.03 ± 0.01 ^d^	0.16 ± 0.01 ^c^	58.6 ± 1.2 ^d^	64.1 ± 1.3 ^e^	49.7 ± 0.11 ^e^	82.2 ± 1.3 ^d^
ChC—0.1%	0.67 ± 0.05 ^c^	71.9 ± 2.2 ^b^	86.6 ± 0.10 ^c^	94.9 ± 0.29 ^d^	96.1 ± 0.06 ^c^	93.1 ± 0.13 ^c^
ChC—0.5%	3.3 ± 0.23 ^b^	92.5 ± 1.1 ^a^	96.6 ± 0.10 ^b^	97.6 ± 0.22 ^b^	97.7 ± 0.17 ^b^	97.4 ± 0.19 ^b^
Asc ac	-	94.0 ± 2.5 ^a^	-	-	-	-
Gentamicin	-	-	99.2 ± 0.10 ^a^	99.1 ± 0.11 ^a^	99.5 ± 0.06 ^a^	99.5 ± 0.06 ^a^

^1^ Values correspond to mean ± standard deviation (*n* = 6 per treatment). GTWE, green tea water extract; ChC—0%, chitosan without GTWE; ChC—0.1%, chitosan with 0.1% of GTWE; ChC—0.5%, chitosan with 0.5% of GTWE. TPHC, total phenolic content; FRSA, free radical scavenging activity; Asc ac, ascorbic acid. Both standards were tested at 25 µg/mL. Different superscripts (a–e) within a column indicate significant differences among treatments (*p* < 0.05).

**Table 2 foods-09-00766-t002:** Effects of treatments and storage time on colour changes in pork chops.

Item	Treatment	Storage Time (Days)					
		0	5	10	15	20	25
L *^1^	T0	52.50 ± 1.93 ^b^	59.02 ± 1.22 ^a^	58.48 ± 1.72 ^a^	61.08 ± 1.28 ^a^	61.03 ± 1.09 ^a^	60.40 ± 1.36 ^a^
	T1	52.31 ± 2.68 ^b^	52.57 ± 2.16 ^b^	54.12 ± 1.64 ^b^	55.54 ± 1.25 ^b^	58.50 ± 1.48 ^a^	58.76 ± 1.98 ^a^
	T2	53.47 ± 2.16 ^b^	54.34 ± 2.31 ^b^	54.27 ± 1.99 ^b^	55.80 ± 1.51 ^b^	55.24 ± 1.38 ^b^	56.04 ± 1.40 ^b^
	T3	54.33 ± 1.85 ^b^	55.79 ± 2.17 ^b^	55.91 ± 0.78 ^b^	56.33 ± 1.54 ^b^	54.50 ± 1.63 ^b^	54.84 ± 1.17 ^b^
a *^1^	T0	2.56 ± 0.43 ^a^	2.05 ± 0.38 ^a^	1.55 ± 0.33 ^b^	1.20 ± 0.29 ^c^	1.22 ± 0.32 ^c^	1.19 ± 0.02 ^c^
	T1	1.76 ± 0.60 ^b^	1.60 ± 0.57 ^b^	1.54 ± 0.43 ^b^	1.25 ± 0.24 ^c^	1.26 ± 0.18 ^c^	1.14 ± 0.49 ^c^
	T2	1.78 ± 0.52 ^b^	1.71 ± 0.36 ^b^	1.65 ± 0.18 ^b^	1.61 ± 0.32 ^b^	1.60 ± 0.18 ^b^	1.61 ± 0.10 ^b^
	T3	1.78 ± 0.36 ^b^	1.70 ± 0.24 ^b^	1.72 ± 0.07 ^b^	1.62 ± 0.22 ^b^	1.60 ± 0.12 ^b^	1.59 ± 0.11 ^b^
b *^1^	T0	11.83 ± 0.58 ^c^	12.78 ± 0.19 ^b,c^	12.07 ± 0.63 ^b,c^	12.69 ± 0.21 ^b,c^	13.30 ± 0.75 ^a,b^	13.84 ± 0.35 ^a^
	T1	10.44 ± 1.01 ^c^	10.10 ± 1.25 ^c^	10.94 ± 1.02 ^c^	11.36 ± 0.78 ^c^	13.13 ± 0.90 ^a,b^	13.22 ± 0.22 ^a^
	T2	10.49 ± 1.21 ^c^	11.22 ± 0.95 ^c^	11.35 ± 1.05 ^c^	11.50 ± 1.04 ^c^	11.46 ± 0.80 ^c^	11.18 ± 0.79 ^c^
	T3	10.51 ± 1.57 ^c^	11.73 ± 0.89 ^c^	11.03 ± 1.34 ^c^	11.12 ± 1.32 ^c^	11.84 ± 0.83 ^c^	11.85 ± 0.65 ^c^
L *^2^	T0	38.60 ± 1.37 ^d^	38.96 ± 0.79 ^d^	43.73 ± 1.39 ^c^	54.37 ± 1.77 ^a^	53.61 ± 1.42 ^a^	52.34 ± 1.55 ^a^
	T1	45.80 ± 1.94 ^c^	43.17 ± 2.25 ^c^	43.69 ± 1.23 ^c^	51.24 ± 0.94 ^b^	54.45 ± 1.19 ^a^	54.27 ± 1.87 ^a^
	T2	43.82 ± 1.35 ^c^	45.04 ± 1.71 ^c^	46.47 ± 2.89 ^b,c^	51.88 ± 1.03 ^b^	49.33 ± 1.22 ^b^	48.13 ± 0.66 ^b^
	T3	43.74 ± 2.32 ^c^	43.70 ± 0.70 ^c^	48.83 ± 0.82 ^b,c^	50.67 ± 1.27 ^b^	50.04 ± 1.75 ^b^	49.65 ± 1.52 ^b^
a *^2^	T0	20.17 ± 1.70 ^a^	22.82 ± 1.49 ^a^	18.21 ± 0.03 ^b^	6.14 ± 1.37 ^e^	6.94 ± 1.16 ^e,f^	5.08 ± 0.46 ^f^
	T1	13.64 ± 0.82 ^c^	12.56 ± 0.87 ^c^	8.76 ± 0.66 ^e^	6.44 ± 0.71 ^e^	6.39 ± 0.55 ^e^	5.05 ± 0.83 ^e,f^
	T2	12.30 ± 0.54 ^c^	11.43 ± 1.73 ^c,d^	11.83 ± 0.37 ^d^	8.23 ± 1.82 ^e^	7.51 ± 0.47 ^e^	6.20 ± 0.52 ^e^
	T3	14.52 ± 0.98 ^c^	13.80 ± 1.51 ^c^	10.88 ± 1.62 ^d^	7.92 ± 0.31 ^e^	7.37 ± 0.53 ^e^	6.76 ± 0.53 ^e^
b *^2^	T0	17.71 ± 1.84 ^a^	17.44 ± 1.55 ^a^	17.19 ± 0.70 ^a^	12.73 ± 0.54 ^b^	10.74 ± 1.52 ^c^	10.35 ± 1.28 ^c^
	T1	17.59 ± 1.56 ^a^	16.11 ± 1.33 ^a^	12.15 ± 0.70 ^b^	9.71 ± 0.72 ^c^	9.91 ± 0.78 ^c^	9.69 ± 0.87 ^c^
	T2	14.29 ± 1.58 ^a,b^	14.26 ± 1.82 ^a,b^	11.09 ± 1.00 ^b,c^	10.91 ± 1.69 ^c^	10.69 ± 0.58 ^c^	10.82 ± 0.30 ^c^
	T3	14.43 ± 1.41 ^a,b^	12.31 ± 1.19 ^b^	11.52 ± 0.95 ^b,c^	10.72 ± 1.87 ^c^	10.79 ± 0.33 ^c^	10.42 ± 1.21 ^c^

Values expressed as mean ± standard deviation. ^*1^, colour of meat surface; ^*2^, colour of bone surface; T0, non-coated chops without antioxidant; T1, chitosan without GTWE; T2, chitosan with 0.1% of GTWE; T3, chitosan with 0.5% of GTWE; Different superscripts (a–f) indicate significant differences among treatment x storage time effects (*p* < 0.05).

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
