# Peer review of "Effects of Chitosan Coating with Green Tea Aqueous Extract on Lipid Oxidation and Microbial Growth in Pork Chops during Chilled Storage"

_foods, 2020, doi:10.3390/foods9060766_

Round 1

Reviewer 1 Report

This study aimed to assess the effects of added chitosan coating alone and in combination with green tea water extract on quality of pork chops during prolonged cold storage. The topic of this study is of relevance, because as mentioned by authors there is an increasing interest in the application of edible coatings like chitosan in food matrices, motivated not only by the increasing consumer demand for safe and stable food products, but also for the rejection of non-biodegradable packaging.

Initials should be avoided in the abstract, such as GTWE, green tea water 186 extract; or FRSA, free-radical scavenging activity .

Sampling and experimental design including four treatments (n= 18 per treatment): T0 (non–coated chops without GTWE); T1 25 (chitosan-coated chops without GTWE); T2 (chitosan-coated chops plus 0.1% of GTWE); and T3 26 (chitosan-coated chops plus 0.5% of GTWE), physicochemical (pH, colour, and lipid oxidation) and microbiological analyses 28 (mesophilic and pyschrotrophic counts) at 0, 5, 10, 15, 20 and 25 days of storage were appropriate. Moreover, statistical treatment of results was correct.

Authors mention in introduction section that “ growth of spoilage bacteria reduces shelf–life and jeopardizes palatability of meat and meat products [6]; thus, the growing demand for pork needs to be attended with an adequate control of LOX and microbial growth [5,6].” However, the impact of chitosan-coated chops plus GTWE was not assed and it can also influence meat palatability. A comment should be included.

There are several studies related with extending the shelf-life of meat and meat products using chitosan coatings, therefore the novelty of this article must be included at the end of introduction section and the major achievements must be highlighted in the conclusions.

Author Response

Manuscript foods-819587

Respect to: Initials should be avoided in the abstract, such as GTWE, green tea water 186 extract; or FRSA, free-radical scavenging activity

Response: indicated abbreviations were removed (FRSA and GTWE) (lines 22-25, 27, 30, 32)

Respect to: Authors mention in introduction section that “ growth of spoilage bacteria reduces shelf–life and jeopardizes palatability of meat and meat products [6]; thus, the growing demand for pork needs to be attended with an adequate control of LOX and microbial growth [5,6].” However, the impact of chitosan-coated chops plus GTWE was not assed and it can also influence meat palatability. A comment should be included. There are several studies related with extending the shelf-life of meat and meat products using chitosan coatings, therefore the novelty of this article must be included at the end of introduction section and the major achievements must be highlighted in the conclusions.

Response: the suggested changes were made (lines 71-80), new references were included so the numbering throughout the text was modified

Note: Modifications were made through the manuscript format, to complement the discussion of the work and adjust to more than 4500 words in the body of text. The abstract was adjusted according to the authors guide.

Reviewer 2 Report

This article aimed to assess the effects of added chitosan coating alone and in combination with green tea water extract (GTWE) on the quality of pork chops during prolonged cold storage. The authors concluded that Chitosan coating with GTWE could be used as an additive for the preservation of pork meat products.

Although the article is not innovative, it contains original and interesting information. 

This article would be improved if the authors clarify or revise the followings:

Line 25. Spell out GTWE to “green tea water extract” since it appears the first time.

Line 39. Remove “,”.

Line 72. Revise “agar” to “broth”. Add “(BHI)”.

Line 73. Remove “from”. Add “(PCA)”.

Line 74. Add “CDMX, Mexico”.

Lines 108-110. Elaborate why the authors chose these bacterial strains for the antimicrobial analysis.

Line 111. Revise to “liquid nutrient (BHI) broth”.

Lines 115-116. Remove “agar”. 

Lines 120-121. How were the authors able to maintain the level of bacteria at < 2.0 log CFU/g?

Line 157. Revise “diluted” to “homogenized for ? min using a stomacher or homogenizer (model, company, and country info)”. Detailed info of the homogenizer used is needed. 

Line 183. Add “among treatments incorporated with GTWE from 0 to 0.5%”.  

Lines 191-193. Rephrase the sentence for clarification, particularly using the term “the highest”. Additionally, the sentence may be revised to “whereas ChC-0.5% (chitosan with 0.5% of GTWE) exhibited the highest inhibitory activity against S. aureus (>90%).”

Line 195. Revise to “a significant (P < 0.05)”.

Lines 198-201. pH values seem to be affected by treatment and storage time for the control sample only, respectively. Therefore, generalizing the findings with the statement “Treatment x storage time interaction on pH values was significant (P < 0.001).” may be misleading. Hence, a revision of the sentence is recommended. The following statement, “At day 25 of storage, the lowest (P < 0.05) pH values were shown in treatments T1, T2 and T3.” may also be misleading since their pH values at day 0 were lower than pH values at day 25. What was the intention of the authors for this statement? This sentence needs revision.

Table 2. I suggest the authors run statistics by storage time (row comparison only) and treatment (column comparison only), respectively, within each item and describe their results accordingly. 

Line 240. Remove “mg”.

Line 295. Revise to “insignificant”.

Author Response

Manuscript foods-819587

Respect to: Spell out GTWE to “green tea water extract” since it appears the first time

Response: the suggested corrections were made in line 23

Respect to: Line 39. Remove “,”.

Response: the suggested corrections were made in line 39

Respect to: Line 72. Revise “agar” to “broth”. Add “(BHI)”.

Response: the suggested corrections were made in line 73

Respect to: Line 73. Remove “from”. Add “(PCA)”.

Response: the suggested corrections were made in line 74

Respect to: Lines 108-110. Elaborate why the authors chose these bacterial strains for the antimicrobial analysis.

Response: the suggested corrections were made in line 109

Respect to: Line 111. Revise to “liquid nutrient (BHI) broth”.

Response: the suggested corrections were made in line 111

Respect to: Lines 115-116. Remove “agar”.

Response: the suggested corrections were made in line 116

Respect to: Line 157. Revise “diluted” to “homogenized for? min using a stomacher or homogenizer (model, company, and country info)”. Detailed info of the homogenizer used is needed.

Response: the suggested corrections were made in lines 158 and 159

Respect to: Line 183. Add “among treatments incorporated with GTWE from 0 to 0.5%”.

Response: the suggested corrections were made in lines 184 and 185

Respect to: Lines 191-193. Rephrase the sentence for clarification, particularly using the term “the highest”. Additionally, the sentence may be revised to “whereas ChC-0.5% (chitosan with 0.5% of GTWE) exhibited the highest inhibitory activity against S. aureus (>90%).”

Response: the suggested corrections were made in line 195

Respect to: Line 195. Revise to “a significant (P < 0.05)”.

Response: the suggested corrections were made in lines 197

Respect to: Lines 198-201. pH values seem to be affected by treatment and storage time for the control sample only, respectively. Therefore, generalizing the findings with the statement “Treatment x storage time interaction on pH values was significant (P < 0.001).” may be misleading. Hence, a revision of the sentence is recommended. The following statement, “At day 25 of storage, the lowest (P < 0.05) pH values were shown in treatments T1, T2 and T3.” may also be misleading since their pH values at day 0 were lower than pH values at day 25. What was the intention of the authors for this statement? This sentence needs revision.

Response: the suggested corrections were made in lines 200-204.

Respect to: Line 240. Remove “mg”.

Response: the suggested corrections were made in lines 243.

Respect to: Line 295. Revise to “insignificant”.

Response: the suggested corrections were made in line 284.

Respect to: Line 74. Add “CDMX, Mexico”.

Response: the suggested correction was made in the manuscript

Respect to: Lines 120-121. How were the authors able to maintain the level of bacteria at < 2.0 log CFU/g?

Response: commonly the initial load in meat 48 h postmortem, is approximately 2-3 log CFU/g. In this study, samples were collected at 24 h postmortem, and were collected in the slaughter plant under acceptable conditions. In other research work carried out by our group, a higher load of bacteria has been reported 3-4.5 log CFU, however, the meat is purchased in the market instead of the slaughter plant.

Respect to: Table 2. I suggest the authors run statistics by storage time (row comparison only) and treatment (column comparison only), respectively, within each item and describe their results accordingly.

Response: In this work we consider executing a 2-way design, because it helps us to know the effect of the factors that explain the behavior of the samples during the shelf life, that is, to know the effect of both the treatment in combination with storage time. In this type of design, the assignment of literals is presented through the rows and columns. This assignment is easy to interpret, because the literals are different, even if we compare between treatments in each sampling day, or when it is interpreted through the storage time for each treatment.
